# Assessing (Social-Ecological) Systems Thinking by Evaluating Cognitive Maps

**Steven Gray** [1,*], **Eleanor J. Sterling** [2], **Payam Aminpour** [1], **Lissy Goralnik** [1], **Alison Singer** [1], **Cynthia Wei** [3], **Sharon Akabas** [4], **Rebecca C. Jordan** [1], **Philippe J. Giabbanelli** [5], **Jennifer Hodbod** [1], **Erin Betley** [2] and **Patricia Norris** [1]

1   Department of Community Sustainability, Michigan State University, East Lansing, MI 48823, USA; aminpour@msu.edu (P.A.); goralnik@msu.edu (L.G.); singer20@msu.edu (A.S.); jordanre@msu.edu (R.C.J.); jhodbod@msu.edu (J.H.); norrisp@msu.edu (P.N.)
2   Center for Biodiversity and Conservation, American Museum of Natural History, New York, NY 10024, USA; sterling@amnh.org (E.J.S.); ebetley@amnh.org (E.B.)
3   Science, Technology, and International Affairs Program, Walsh School of Foreign Service, Georgetown University, Washington, DC 20057, USA; cynthia.wei@georgetown.edu
4   Institute of Human Nutrition at Columbia University, New York, NY 10032, USA; sa109@columbia.edu
5   Department of Computer Science and Software Engineering at Miami University, Oxford, OH 45056, USA; giabbapj@miamioh.edu
*   Correspondence: stevenallangray@gmail.com; Tel.: +646-915-2915

**Abstract:** Systems thinking (ST) skills are often the foundation of sustainability science curricula. Though ST skill sets are used as a basic approach to reasoning about complex environmental problems, there are gaps in our understanding regarding the best ways to promote and assess ST learning in classrooms. Since ST learning provides Science, Technology, Engineering, and Mathematics (STEM) students' important skills and awareness to participate in environmental problem-solving, addressing these gaps is an important STEM learning contribution. We have created guidelines for teaching and measuring ST skills derived from a hybrid of a literature review and through case study data collection. Our approach is based on semi-quantitative cognitive mapping techniques meant to support deep reasoning about the complexities of social–ecological issues. We begin by arguing that ST should be evaluated on a continuum of understanding rather than a binary of correct/incorrect or present/absent. We then suggest four fundamental dimensions of teaching and evaluating ST which include: (1) system structure, (2) system function, (3) identification of leverage points for change, and (4) trade-off analysis. Finally, we use a case study to show how these ideas can be assessed through cognitive maps to help students develop deep system understanding and the capacity to propose innovative solutions to sustainability problems.

**Keywords:** social–ecological systems; cognitive mapping; sustainability education; sustainability science; leverage points

## 1. Introduction

Addressing contemporary social–ecological problems in an increasingly complex world requires that the next generation of sustainability scientists possess "systems thinking" (ST) skills. The importance of developing these ST skills is reflected across many sustainability-related educational programs. For example, many interdisciplinary environmental (IE) degree programs, the goal of which is to prepare students to become "sustainability-oriented scientists, leaders, problem-solvers, and decision makers" [1], have identified ST as a fundamental requirement [2]. Further, attempts to define core competencies for students in these interdisciplinary and diverse environmental fields

have identified ST skills as an important element [3–5]. ST thinking is a core Science, Technology, Engineering, and Mathematics (STEM) skill related to learner capacity to engage "wicked problems" and participate meaningfully in identifying pathways for positive change. Wicked problems are those with high levels of uncertainty and value conflict where there is no single solution to the problem (see [6] for more details about "wicked problems"). With this definition, nested and complex social–ecological problems at the core of most sustainability issues are instances of wicked problems.

Focusing on social–ecological problem-solving using an ST approach results in three primary benefits. First, ST requires students to both identify particular dynamics and relationships of actors and mechanisms in a system and, also, to step back and analyze the dynamics of the system as a whole [7]. When dealing with complex or wicked problems, this alone is a useful step in student engagement and problem-solving capacity. Often the scope and scale of sustainability problems, as well as the inherent value dynamics, which elicit emotional engagement, can be overwhelming for learners [8]. This can lead to apathy or powerlessness, which are both impediments to meaningful participation and also learning [9]. The ability to identify and define components and understand the dynamics of a system in a systematic way can contribute to learner engagement with sustainability issues. Second, if learners can think critically about the complex dynamics of a system, they are better prepared to predict a system's behavior, engineer more favorable outcomes (see identifying "leverage points" discussed by Meadows [10]), and evaluate the trade-offs between different decisions made within the system. Because human and ecological well-being are interdependent, ST skills can enable students to develop better ways to reason about possible system outcomes and suggest management plans that anticipate trade-offs to minimize negative impacts and improve both ecosystem health and human well-being. Third, a benefit of fostering ST skills is that they can play a role in facilitating integrative reasoning across social and natural scientific disciplines. For example, given the interdisciplinary nature of complex problems such as global climate change or the potential impacts of genetically modified crops, ST can engender research and problem-solving that draw on different ecological and social principles that routinely characterize contemporary sustainability problems.

However, while many recognize the value of ST, considerable gaps remain in understanding how to teach and measure ST [11]. The available materials for educators are non-standardized and largely ad hoc [11,12] with some notable exceptions emerging (see [13]). Below we suggest guidelines for ST assessment in sustainability that rely first on thinking about ST on a continuum of student understanding and on iterative evaluation to improve our understanding of student ST learning over time. We then suggest four fundamental dimensions of ST that provide a framework for understanding degrees of ST, which include evaluating student understanding of: (1) system structure, (2) system function, (3) identification and negotiation of leverage points for change, and (4) trade-off analysis (Table 1). We conclude with an example ST assessment using a semi-quantitative cognitive mapping technique we have used in a variety of STEM classes that touch on different contemporary sustainability issues.

*Systems Thinking on a Continuum*

To improve ST teaching and learning, we suggest educators view ST on a continuum of development rather than as a binary of correct/incorrect or present/absent at one point in time. Recent research suggests that ST is constructed over time as new information is obtained and new connections across information are made [14]. Therefore, to understand the development of ST, we should measure it repeatedly, over time within a classroom or within a curriculum and adapt instruction to specific challenges that student's face as they represent their understandings of a system. Further, some have suggested that degrees of ST may have explicit stages in a progression, similar to a learning progression [15], although these ideas remain untested. Below we propose four dimensions of ST that instructors may use to evaluate and provide feedback on student understanding of complex social–ecological issues confronted in the science classroom.

**Table 1.** Four elements of systems thinking (ST).

| Components of ST | Learning Outcomes | Level of ST Sophistication |
|---|---|---|
| Structure | Recognize and define basic systems language, properties, and behaviors of basic systems. | Low–Medium |
| | Identify and explain system archetypes and be able to explain their impacts. | |
| | Analyze system components—social and ecological elements, their connections, slow and fast drivers, endogenous and exogenous drivers, and scales above and below. | |
| | Apply systems thinking/mapping to explain the interconnectedness of human and natural systems. | |
| Function | Identify functions within a system, anthropocentric and not. | Medium–High |
| | Evaluate how changing function influences other system structures. | |
| Leverage points | Apply systems thinking/mapping to generate multiple scenarios to inform a decision-making process around human and natural systems. | High |
| | Practice anticipation—i.e., forward thinking in research and engagement—in addition to reaction. | |
| | Identify leverage points for changes in management. | |
| Trade-offs | Explain that any sustainability issue both influences and is influenced by multiple spatial and temporal scales given the interconnectedness of social–ecological systems. | Highest |
| | Recognize that systems are nested and explain cross-scale dynamics. | |
| | Analyze the trade-offs a management decision or exogenous shock may result in within a social–ecological system, and at the scales above and below. | |

## 2. Teaching and Evaluating Four Dimensions of Systems Thinking

### 2.1. System Structure

The structural aspect of ST can be evaluated fairly straightforwardly and provides insight into how learners 'see' or make sense of a system's boundaries and its composition. While there is generally no right or a wrong system structure, there are robust indicators of a functioning system that rely on a mix of logic, evidence-based principles uncovered by previous scientific research, and conceptual understanding. By identifying the conceptual boundaries of a system, the composition of a system, and the relationships between elements, including feedbacks between elements within a system [10], learners can define the structure of a system, which is considered the basis for higher-order reasoning about complex systems [16]. The observational and conceptual recognition of structures has been linked to nearly every mode of inquiry and discovery in science, philosophy, and art [17]. The relationships between structures are what influence system behavior and give a system its shape, which can be hierarchical or networked. Understanding structures is analogous to the "whats" of the system and how relationships (e.g., stocks, flows, and feedbacks, see [10]) between these "whats" are defined [18].

### 2.2. System Function

Functions are the outcomes of the system, based on the types of structures in the system and the relationships between them. Therefore, system function represents a higher-order teaching and learning about ST than simply representing its structure. It is important to note that in real-world systems, functions are dynamic and can only be evaluated by measuring changes in quantity or quality of structural elements at certain snapshots in time which often requires dynamic system representations

such as discrete-event simulation or higher-level system dynamics and agent-based models [19], or alternatively asking students qualitative questions to uncover their thinking about systems operations. In biological terms, the characteristic exchanges and processes within an ecosystem that allow its continuation are its functions, including energy and nutrient exchanges, regulation of climate and hydrological cycles, and decomposition and production of biomass [20]. In technological systems, functions may be engineered, such as whether communication or information networks are operable. In social terms, the function of institutions or organizations may be to develop rules or laws that maintain a quality of life or social stability. Understanding many sustainability issues, such as energy transitions for example, requires understanding how biological, technological, and social systems function in interaction with each other to produce outcomes that are more than the sum of their individual parts.

*2.3. Leverage Points*

Leverage points, as described by ST scholars, are key places within a system that can be reasonably changed to steer systems toward a preferred state [10]. It is only after the structure and function of systems are sufficiently understood that students can reasonably hypothesize and test ideas about altering system dynamics with a goal or preferred state in mind [21]. At the highest level of ST, understanding structure and function sufficiently to be able to change a system's trajectory, or structures within it, eludes even some of the greatest scholars, given the uncertain and complex nature of the systems in which many 'wicked problems' are embedded. Still, teaching students to identify opportune places to intervene in a system to begin to address persistent problems is an integral component of ST-driven problem-solving. This is an empowered stage of the ST continuum, as well as an important part of sustainability learning and practice.

*2.4. Trade-Offs*

A final, critical element of ST is the understanding that any change to system management will always result in changes to other system structures and functions—i.e., trade-offs. Trade-offs occur when one function is reduced as a consequence of another being increased. In some cases, a trade-off may be an explicit choice, but in others, trade-offs arise without intention or even awareness that they are taking place [22]. This is the general nature of wicked problems [23] and, therefore, an important part of learning about and for sustainability. To avoid unintentional trade-offs and avoid adverse effects when trying to achieve favorable outcomes for both human and environmental functions, learners must thoroughly understand the relationships between system structures and their functions.

## 3. Assessing Systems Thinking by Evaluating Cognitive Maps

Here we discuss an approach to evaluating learner progression on a ST continuum using a cognitive mapping technique (Figure 1). This activity involves asking students to develop cognitive maps that identify concepts/ideas as nodes and causal relationships between concepts as links connecting these nodes that they describe qualitatively (e.g., [24]). Such cognitive maps are considered external representations of individually held "mental models" about a real-world complex system, with the structure of these mental models expected to correspond to the student's perceived structure about a system [25].

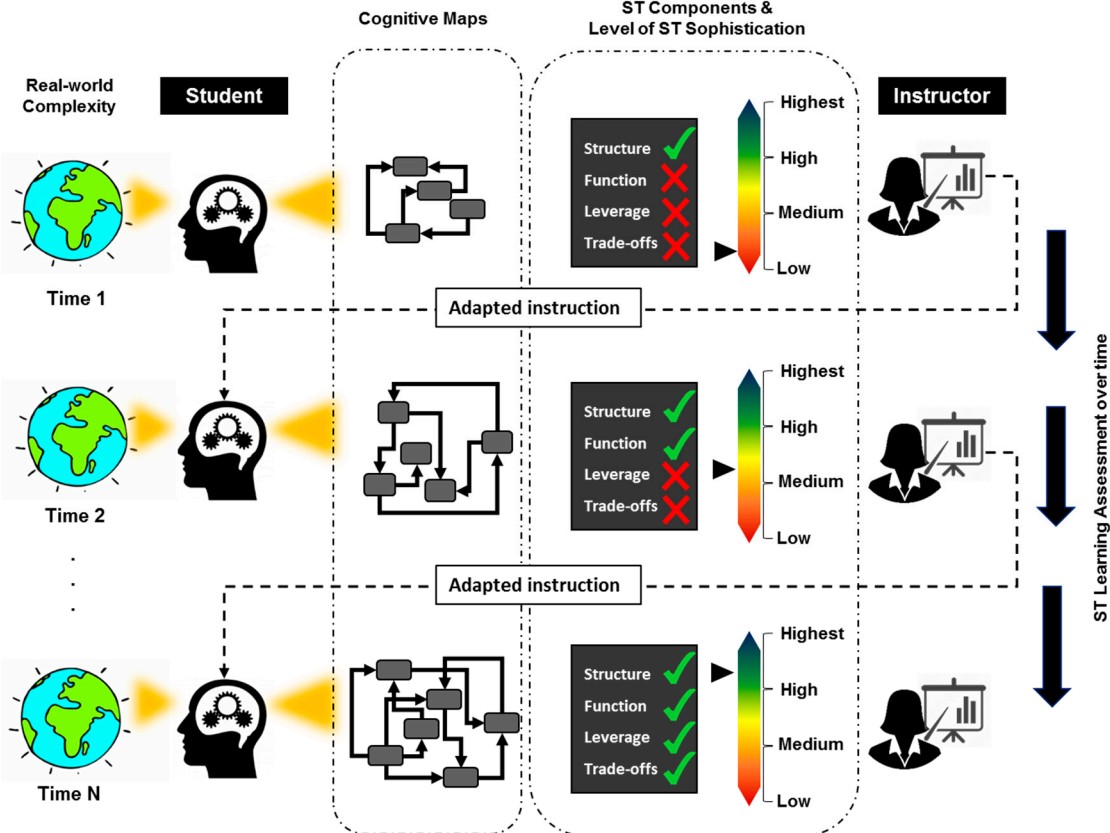

**Figure 1.** Development of students ST learning through repeated measurement over time within a classroom or within a curriculum and adapted instruction based on evaluating learner progression on a ST continuum.

Student cognitive maps are typically assessed for: (a) degree of richness, (b) networked structure of ideas, and (c) overall degree of explanation present in the map [18,26]. Most of these maps, however, assess an individual's ST in a static fashion, relying on useful but somewhat limited qualitative explanations as evidence that students are moving from linear and disjointed ideas to more networked and complex ideas.

As suggested above, however, ST assessment needs to evaluate conceptual changes (i.e., learning) over time, rather than at a fixed place in understanding. This has been done in a limited fashion with the same individual through multiple measurements during a period of time to evaluate how quantitative (e.g., number of concepts and connections) and qualitative (e.g., relative 'importance' of different components) representations of understanding change over time in learners' cognitive maps [7]. In another example, when teaching the case study, Designing an Urban Green Infrastructure Network: Balancing Biodiversity and Stakeholder Needs [27], the authors compared student cognitive maps of a university campus as a social–ecological system created before and after the case study activity as a way to assess changes in student understanding. These authors observed changes in the degree to which these systems were coupled, from a relatively simplistic view of the social elements and natural elements being represented separately to a more integrated view as indicated through increased connections between natural and social elements.

In a larger study, Dauer et al. described changes in student-constructed "gene-to-evolution" cognitive maps and explanations over a semester of an introductory biology course [28], and Dauer and Long characterized changes in student thinking and explanation quality 2.5 years after the introductory biology course [14]. In the longitudinal study, the authors identified distinct groups of learners by combining analysis of the cognitive maps with other formal written and verbal assessments to

characterize changes in student thinking over time. The completeness of student cognitive maps varied with the quality of their explanations about the relationship between content covered in the course. The studies above indicate a potential for cognitive mapping to not only help students attain conceptual learning goals but also to develop ST in ways instructors can measure students' ST skills and, thus, improve their teaching.

*Case Study of Teaching Social–Ecological (Systems) Thinking*

Students in an introductory sustainability science class at a large research university were provided a scientific article and several popular periodical artless about the intersection of climate change and terrorist activity and how drought is influencing social conflict and change (see [29]). Students in the class (N = 40) represented mostly lower-level undergraduates (freshman and sophomores) enrolled in the course. Roughly a third of the students were STEM majors and about half of those majors were enrolled in majoring in an environmental studies program. They were then asked to develop cognitive maps that reflect their understanding of the scientific and popular articles. Applying our four tenets of ST to representative maps developed during this assignment provides insight into the continuum of ST across students. Using the same or an expanded assignment over the course of the semester would also allow insight into the continuum experienced by an individual student over time. Figure 2 shows three cognitive maps representing student understanding of the concepts and relationships between climate change and terrorism activity based on synthesizing the material provided. Cognitive maps from the same class discussed above were independently ranked as representing high, medium, and low levels of ST by 10 university faculty who study ST but with no formal guiding criteria for what should be considered evidence of ST. Figure 2 shows one example from each ranking category. Structural network metrics (e.g., number of concepts, connections, centrality of concepts, density, etc.) applied to the student maps indicate general patterns on a continuum of ST, with higher ST assessment associated with more concepts and connections between concepts, higher density, but overall lower levels of the ratio of number of connections/number of concepts. The relative importance of different concepts included also varied, with the highest ST map representing "desert encroachment" and "climate change" as the most central variables, with medium ST representing "violence" and "terrorism," and the lowest ST representing "drought" and "natural resources" as most central to the map indicating differences in the degree of conceptual specificity and scale decreases as student ST decreases. These data suggest potential norms for higher and lower forms of ST and identify trends or progressions in conceptual focus on different structures that comprise systems and influence system function.

The first step of ST assessment should start with questions about *structure*, including: (1) presence/absence of necessary elements and relationships, (2) presence/absence of unnecessary structures, and (3) theoretical or empirical appropriateness of the relationships between these elements, based on current scientific understanding of the system. This step requires the instructor to have either well-developed knowledge about the study system or a framework of components already developed.

The second step of ST assessment addresses function. This step is not quite as straightforward, but there are tools and examples to facilitate the process. Given that function is highly dependent on system structure, however, how students understand the relationship between structure and function is also important for instructors. The latter can be more difficult. While some studies (e.g., [18]) have used qualitative descriptions about the relationships between variables to assess student understanding of causal elements and emergent properties of systems, these rubrics do not often allow students to test ideas, change model structure iteratively, and reason through 'what if' scenarios to fully appreciate the interdependencies of how structure relates to function. Further, while computational systems modeling platforms used by environmental scientists and in graduate training programs do provide this type of reflective evaluation (see Systems Dynamics and Agent-based Modeling), these software packages require considerable training and modeling skills.

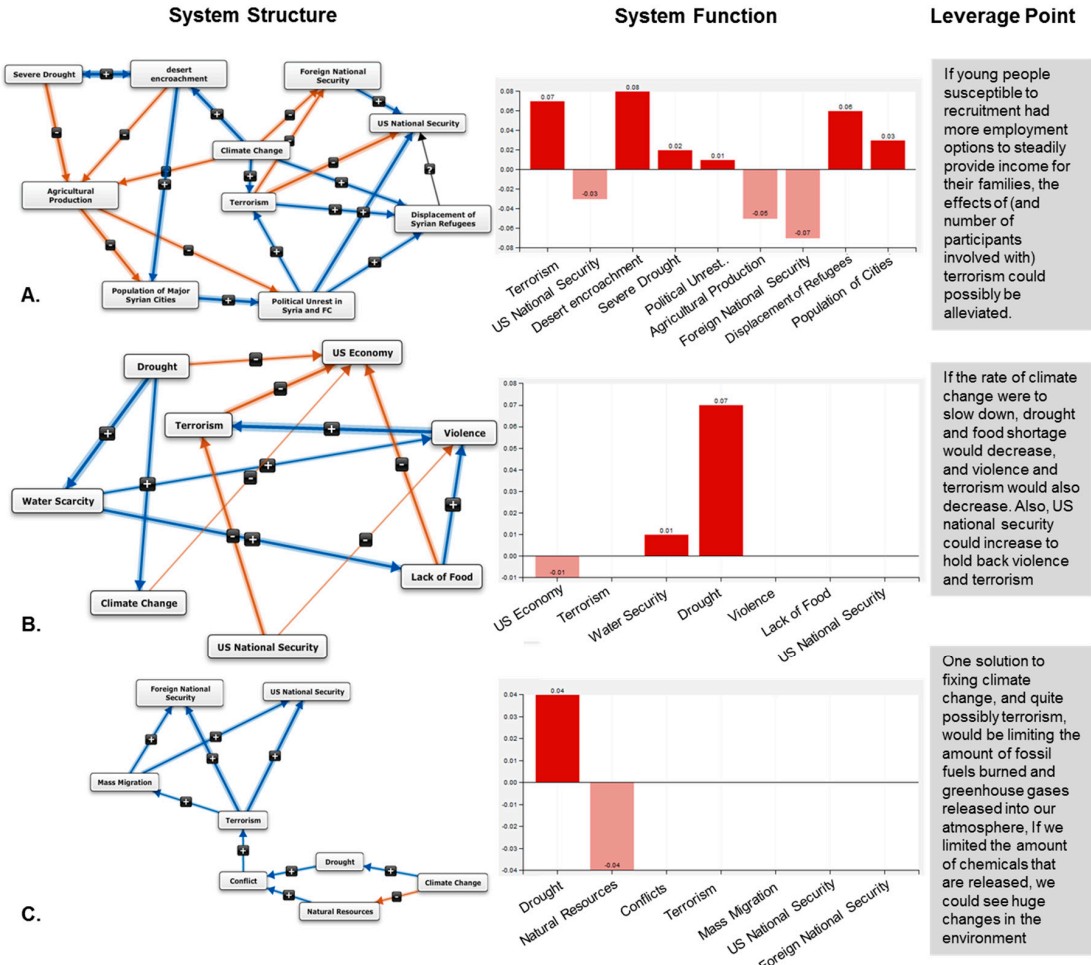

**Figure 2.** Student cognitive maps of system structures that represent understanding of links between climate change, natural resource availability, and terrorism as outlined by research by Kelley et al. [29] (2015) using the free online software www.mentalmodeler.org. Cognitive maps include elements that can increase or decrease in quality and quantity and relationships between elements are represented by positive influences (blue lines) and negative influences (red lines). Cognitive maps were ranked independently by 10 university instructors indicating that (**A**) represented high ST, (**B**) represented medium ST, and (**C**) represented low ST.

More recently, though, some researchers have designed ST tools that rely on semi-quantification of qualitative associations to draw on the strengths of the easy-to-use qualitative cognitive mapping (e.g., www.mentalmodeler.org; an online fuzzy cognitive mapping tool) [30]. Fuzzy cognitive mapping (FCM) [31] is a certain semi-quantitative cognitive mapping technique. FCMs represent systems as directed and weighted graphs (Figure 2), where the nodes of the graph qualitatively represent elements of the system (i.e., concepts), and the edges between the nodes quantitatively represent the direction and strength of causal relationships between concepts. Consequently, FCM serves as a cognitive mapping activity where higher ST components become apparent. Unlike qualitative cognitive maps, FCM enables students to represent a system's dynamic behavior (i.e., *function*) through a simulation of causal propagations in discrete time-steps [19]. These semi-quantitative cognitive maps provide students a way to test ideas related to both structure and function and to see whether model function can be compared to empirically generated data [7,30,32,33]. Such tools also provide ways for STEM educators to assess how well students use ST and provide directed feedback for improvements, so that function and structure can be evaluated in tandem by students and teachers (Figure 1). An added

benefit of using these tools as a feedback platform is that the integrated assessment can provide an additional opportunity for student learning about ST process and metacognition development.

The third step of assessing ST, which is difficult in a qualitative cognitive map alone, is to assess an understanding of leverage points. Pairing the cognitive maps with student writing, though, provides a strategy to evaluate: (a) identification of leverage points and (b) sophistication of thinking. For example, the student essay associated with the cognitive map that we rated as the most sophisticated in Figure 2A indicated that leverage points to decrease terrorism activity included an increase of employment opportunities that might reduce the number of individuals susceptible to terrorist recruitment. The other two maps, on the other hand, focused on higher-order leverage points of reducing the rate of climate change, which is arguably a more difficult point of intervention (but still an open area of debate; see discussions about climate change mitigation versus adaptation in [34]). Discussion of identified leverage points can provide insight into mechanisms or hypotheses for change, leading to policy developments (e.g., addressing terrorism) or experiments (e.g., manipulating variables) useful for developing deeper understanding.

However, the ultimate goal of assessing ST in the classroom is not to use leverage points to precisely engineer a system toward a particular preferred state (e.g., determining the efficient use of water for balancing green water (GW) and blue water (BW) use for sustainable crop production and achieving higher food security [35]). Rather the goal is for students to develop fluency with ST in ways they can apply to the evaluation and address of other problems in the future. This capacity is developed through the ability and agency to propose creative ideas based on deep understanding of a system, which can then lead to innovation about policy interventions or problem-solving initiatives and/or informed discussions about hard and seemingly intractable sustainability problems in the classroom. Being able to engage in informed dialogue about system dynamics and change can prepare STEM students to participate in collaborative sustainability problem-solving in the future.

The teaching case study, Using Systems Maps to Analyze Socio-Environmental Issues: A Case Study of Geoduck Aquaculture in the Puget Sound [36], reflects this process and goal. In this publicly accessible resource, the authors have designed a series of activities aimed at guiding students from learning about the problem of geoduck aquaculture to identifying leverage points and evaluating proposed policy solutions. Students begin by developing a systems map, and then use the systems map to identify leverage points in the system. With these leverage points identified, the students then pose potential problems and solutions and, finally, evaluate proposed policy solutions. The assessment of student ST skills through these activities focuses not on arrival at a "correct" answer, but rather the development of their understanding of the system and their reasoning to identify leverage points and evaluate possible solutions.

This highlights another teaching and learning goal. With respect to student learning, the key outcome is to understand that social–ecological issues rarely have a single correct answer given their complex interconnections; that is, actions taken today can have different impacts on different users within a system and unintended consequences not only in the system under study but in other geographical locations and in future time periods. While it may be argued that systems can be managed for a single function and undesired effects dealt with as they arise, a sustainability perspective would advocate for minimal negative impacts on human or environmental well-being. Thus, an awareness of connections within a system through ST manages for minimal adverse trade-offs from the beginning. However, again, it is not the intention that students can then precisely engineer a system toward a particular preferred state with minimal adverse trade-offs, but that they gain the skills in ST to assess the broader impacts of decisions on sustainability at multiple spatial scales and recognize that management actions will have outcomes on long timescales. In total, cognitive maps, and possibly scenario outputs, paired with student writing can be used to assess student understanding of trade-offs.

## 4. Conclusions

*Focusing on Assessment to Determine Best Practice for Teaching Systems Thinking*

Although ST skills are widely believed to be a unifying framework for many STEM disciplines [37] and are critical for handling the complexity facing the world in the coming decades, assessing and teaching ST in the classroom is far from becoming a routine part of classroom practice. Developing ST skills requires students to integrate concepts within and across disciplines and coursework [38] and to understand the networked structure of these concepts. This helps students to better understand how social and ecological systems are coupled and influence one another. If ST thinking is going to be used in practice, then instructors need to identify ST learning goals when designing ST curriculum around a particular context. The framework we suggest here provides a structure for formulating questions that help instructors achieve these goals: *What evidence demonstrates that students have mastered structural understanding of a system? What evidence shows that students have sufficient understanding of system function? What evidence suggests that students understand how to intervene and change the dynamics of systems? And what evidence shows that students have adequate understanding of broader impacts of decisions at multiple spatial and temporal scales?*

The ability of students to engage in ST sufficiently to transfer these ST skills to creating innovative solutions around complex and 'wicked problems' may be the biggest contribution of ST in formal STEM and sustainability learning environments. ST is a highly transdisciplinary, synthetic, and generalizable construct, and, therefore, it is also considered as a useful way for students to integrate and synthesize knowledge across domains. Such systemic thinking generates scientific habits of mind that are useful frameworks for reasoning and abstracting about a range of systems that underlie many contemporary problems. We have offered suggestions for concrete ways to assess student ST development through the use and interpretation of cognitive maps. Evaluating system structure and function, evaluating trade-offs, and identifying leverage points are tangible ways for students to express their systems thinking abilities and for educators to evaluate those abilities. As we move toward more sustainable, holistic, interdisciplinary approaches to wicked problems, ST skills will help us solve them.

**Author Contributions:** Conceptualization, S.G., E.J.S., E.B., C.W., S.A., A.S., P.J.G., R.C.J., J.H., P.N., L.G. and P.A.; Data curation, E.B.; Formal analysis, S.G.; Funding acquisition, S.G., E.J.S., P.J.G. and R.C.J.; Investigation, E.B. and C.W.; Methodology, A.S. and P.J.G.; Supervision, E.B.; Writing—original draft, S.G., E.J.S., E.B., C.W., A.S., P.J.G., R.C.J., J.H., P.N. and L.G.; Writing—review & editing, E.B., C.W., S.A., A.S., J.H. and P.N.

**Funding:** This research was funded by the National Science Foundation (award number DUE-1711260 and 1711411).

**Conflicts of Interest:** The authors declare no conflict of interest. The funders had no role in the design of the study; in the collection, analyses, or interpretation of data; in the writing of the manuscript; and in the decision to publish the results.

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
