# Peer review of "Assessing (Social-Ecological) Systems Thinking by Evaluating Cognitive Maps"

_sustainability, doi:10.3390/su11205753_

Round 1

Reviewer 1 Report

This paper presents a framework structured in four levels (namely, structure, function, leverage points and trade-offs) for understanding and assessing System Thinking (ST). These four levels are a quite rather smart way of conceptualizing and arranging ST, and could potentially be very useful for planning instruction along a progression.

However, I have some doubts or concerns about the content of the manuscript

The title of the paper talks about “teaching”. Why then so much focus on the evaluation? As said in the first subtitle under “conclusions”, focussing on evaluation helps determine the best strategy for teaching, but there is not any specific suggestion about the way ST should be taught, apart from a vague recommendation to use Concept Maps. The actual theme of the paper is a theoretical proposal about how to scaffold ST (until line 146); the rest is not centrally connected.

I suggest the authors they try to be more explicit about the real purpose and affordances of the article.

In fact, the reader may have problems to understand what’s being assessed. Are ST and Concept maps equivalent concepts? Line 122 reads “the first step of ST assessment”, but the proposed measures are usually used to evaluate CM. I think this should be clarified, and that CMs are only one of the possible approaches for developing and measuring ST, or otherwise explaining more clearly why they feel allowed to do such a strong equivalence.

 “Concept maps” are defined in reference to the heuristic tool developed or at least made popular by John Novak (Cornell University). Concept maps are maps of concepts connected by linking words to form prepositions, and all the propositions referred to a concept make its meaning. But so defined CM have always a clear hierarchical structure. I.e., in a concept map the concepts should be represented in a hierarchical fashion with the most inclusive, most general concepts at the top of the map and the more specific, less general concepts arranged hierarchically below. Often this most general concept takes the form of a question which contributes the meaningful context.

In that sense, the images in the text don’t correspond to concept maps. It might be enough to explain why they take a different form or even to choose another word, such as “mind map”, which is less constrained.

 Figure 3 is difficult to understand. The labels of the graphs are imposible to read, and it’s no clear which information are they trying to convey. Either the authors include a larger image, with higher resolution, where the labels are visible, or they show only a part of the image, that can thus be shown larger and easier to read, or improve the description in the bottom.

 Lines 238-9: More recently, though, some researchers have designed ST tools that rely on semi-quantification of qualitative associations to draw on the strengths of qualitative concept mapping. – It would be good to include some references to these tools.

Lines 306-7: The framework we suggest here provides a structure for answering these questions.

I would say the framework (the idea of the four sequential levels to assess ST) help you formulate these questions (i.e., know better what to look for in the evaluation tool; e.g. What evidence demonstrates that students have mastered structural understanding of a system?), but no necessarily solve them.

 Line 310: I agree that the ability to engage in ST thinking is crucial for formal ESD; this has been somewhat developed in the introduction. But which could be the link with STEM? I invite the authors to make it a bit clearer in the text.

Author Response

Please find attached our responses to reviewer 1

Reviewer 2 Report

Thank you for sharing your MS. I enjoyed reading it and learning more about your approach to teaching and assessing systems thinking. I do have a few questions and some concerns that if attended to would strengthen your paper.

wicked problems - This needs more background when you first introduce the term and its usage. Perhaps a footnote? Also, you need a reference here. One comes along in the fourth time you use the term but it should be upfront and in your intro. 

standardization, etc. - I am wary of looking to standardize materials and processes for learning about and assessing ST. You note in your paper that systems rarely hold a single correct answer in their interconnections and that there are different impacts on different users. This all mirrors the situatedness of systems and related issues. Does it make sense to standardize our teaching and assessment of what is inherently contextual?

QUAN and QUAL - you note at a few points that the QUAL data has limitations and is strengthened with quantification. Why not include both? There are benefits to both, and particularly including QUAL in assessing changes along a continuum.

boundaries, categories, levels, etc. - I do find the terms problematic in teaching and assessing systems thinking. You identify that relationships are inherent in systems - elements cannot be teased apart from those that they are connected to. Also, that continuum again. Does the teaching and assessment (and boundaries, categories, levels, etc.) actually reflect the qualities and complexity of systems thinking itself? 

please do a careful review of writing to check for errors (e.g., 's for plural) and repetition of terms in consecutive sentences (e.g., overwhelm).

Author Response

Please find attached our responses to reviewer 2.

Reviewer 3 Report

Dear authors,

Thanks for submitting to Sustainability. Please find my comments in the attached document. Hope all these comments will help you to prepare an improved version of your manuscript.

Author Response

Please find attached our responses to reviewer 3.

Round 2

Reviewer 1 Report

The manuscript has been significantly improved. At least, it's much better focused and its real affordances are more clearly recognized. The authors are right in changing the focus from "teaching" to monitoring the progress in ST skills using (and evaluating) cognitive maps.

Figure 2 is still unreadable and has thus low value. I insist the captions should be made more visible, by enlarging the image or showing only one of the three cases, as an example. Also, the added value of Figure 1 is limited.

While the framework and the suggestions for STEM teachers are right and have unquestionable value for the teaching and assessment of ST skills, the case study is quite obscure. Neither the method nor the quantitative results are clearly shown. To provide a few examples, what are the results of the evaluation by these 10 faculty members, how many maps were placed as low, medium, high quality? Albeit considering it a continuum, how are these categories defined?,and many other questions. Even if it's clear the main purpose of the manuscript is not to analyze the outcome of this learning activity, being more transparent about the details of the case study would make the conclusions you draw more reliable.

Other minor points include the excessive repetition of the word "system" in lines 47-66.

Author Response

We appreciate these comments and have changed the manuscript (and figures and description of the case study) in line with the reviewer. We thank the reviewer for their time and comments which improved the manuscript.

Reviewer 3 Report

Dear authors,

Thanks for submitting a revisited version of your paper. It is fine by me now. 

All the best,

Author Response

THanks!